# RealLife-Bench: A Benchmark for VLMs on Open-Ended Daily Activities with Holistic Evaluation

## Abstract

We introduce **RealLife-Bench**, a holistic evaluation framework designed to rigorously assess Vision Language Models in open ended household tasks. First to simulate realistic daily activities we manually annotate scene agnostic task skeletons and employ Hierarchical Topological Projection to map these high level semantic instructions such as "I am hungry" and their corresponding multi-solution directed-acyclic-graphs onto diverse simulation environments to generate executable task completion trajectories. Second we establish a comprehensive evaluation framework comprising an end to end online evaluation and five distinct process evaluations to diagnose specific capability bottlenecks. RealLife Bench comprises 1,021 online evaluation episodes and over 10,000 offline process evaluation samples. Our extensive experiments reveal the correlation between distinct capabilities and the completion of embodied tasks identifying perception grounding as the most critical success factor for current VLMs. We will make the code and dataset publicly available.

## 1. Introduction

Embodied intelligence systems integrated with Multi-modal Large Language Models (VLMs) have demonstrated significant advantages in generalization and long-horizon planning, gradually becoming the mainstream solution for embodied AI. However, existing works struggle to effectively guide VLMs toward better application in embodied tasks. For instance, benchmarks like RoboVQA(Sermanet et al., 2024) and ERQA(Gemini Robotics Team et al., 2025) treat embodied tasks as static visual question answer (VQA) tasks, neglecting the assessment of the model's ability to interact with the environment as an agent. Benchmarks like

LIBERO(Liu et al., 2023) and Calvin(Mees et al., 2022) allow Vision-Language-Action models (VLAs) to interact with the environment through fine-grained primitive actions; However, the learning of primitive actions is still in its nascent stage, thereby masking both the capabilities and the limitations of VLMs. Embodied Reasoner(Zhang et al., 2025b) focuses on task planning, allowing VLMs interact with physical environment via predefined semantic actions, and utilizes task success rate as a metric to guide the improvement of VLMs. However, VLMs offer more than just task-planning abilities; they also provide broad visual perception, the capacity to infer intentions, and extensive world knowledge acquired from large-scale multimodal datasets. Besides, relying solely on task success rates makes it difficult to directly guide VLM , and the need for manual case analysis reduces iteration efficiency. Furthermore, the task instructions used in current benchmarks are still constrained to idealized experimental settings that differ substantially from real-world applications. For example, existing data often uses explicit instructions like "pick up the apple and put in the plate," whereas a real user might simply say, "I want some fruits." This gap ignores the essential capabilities required of VLMs as the foundation for VLA systems.

To address these issues, we introduce **RealLife-Bench**, a benchmark centered on real user needs for VLMs in embodied tasks, using household scenarios as a case study to guide the iteration of base VLM models. Specifically, we manually designed instructions covering different levels of understanding difficulty, along with corresponding completion logic represented by graphs. These constitute scene-agnostic task skeletons. Each node is annotated with diverse expansion conditions and the abstract actions required for transitions between nodes. This structure flexibly supports multi-solution tasks and possesses high scalability. We use hierarchical topological projection (HTP) to project these task skeletons onto specific scenes, fusing instructions with the environment to generate data for online evaluation. Subsequently, by reproducing the abstract actions required for node transitions, we project this data into action sequence trajectories capable of completing the tasks.

We built an end-to-end (E2E) online evaluation based on the logical loop of Observation-Execution-Feedback. VLMs are required to execute instructions based on first-person ob-

[1]Anonymous Institution, Anonymous City, Anonymous Region, Anonymous Country. Correspondence to: Anonymous Author <anon.email@domain.com>.

Preliminary work. Under review by the International Conference on Machine Learning (ICML). Do not distribute.

servation history by outputting the target object's bounding box (BBox), a mechanism that mirrors biological perception-action loops. Complementing this interaction, precise feedback is provided to facilitate the alignment of the VLM's internal knowledge with the environment, thereby enabling effective reflection. Considering the accumulated error in E2E online evaluation, RealLife-Bench combines annotated data with trajectories generated during online evaluation to include five process evaluations decoupled from the E2E online evaluation: task decomposition, progress management, action prediction, perception grounding, and feedback correction.

Through extensive experiments, we found that Gemini 3.0 Flash achieves only a 52.6% success rate on RealLife-Bench E2E evaluation, indicating that mainstream VLMs still struggle to competently handle challenges that closely resemble real-world applications. The process evaluations assess VLM performance across five capability dimensions, revealing the correlation between these capabilities and online performance, as well as potential directions for improvement.

Our contributions can be summarized as follows:

- We introduce RealLife-Bench, a benchmark encompassing 15 daily activities across 120 scenes, totaling 1021 tasks. By designing instructions with varying levels of difficulty for each task, we aim to guide VLM toward the capabilities demanded by real-world applications;

- We redefine the action interaction paradigm and incorporate a feedback module to fully leverage the reasoning and perception capabilities of VLMs during environmental interaction. We establish an automated dynamic trajectory evaluation scheme, which balances the freedom of environmental exploration with the efficiency of automated scoring;

- We establish a holistic online evaluation pipeline for VLM-based embodied tasks and construct process evaluations by decomposing embodied tasks into key capability, revealing critical capabilities and influencing factors in embodied task completion;

- Extensive experiments demonstrate that the semantic understanding difficulty of instructions impacts task success, and that deficiencies in perception and grounding serve as the primary bottleneck constraining VLMs in embodied tasks.

## 2. Related Work

### 2.1. Interaction between VLMs and Environments

Numerous studies have investigated VLM performance across various embodied capabilities, including spatial relationship unerstanding, object attribute recognition, and long-horizon task planning (Jiang et al., 2025; Majumdar et al., 2024; Zhao et al., 2025; Windecker et al., 2025; Chow et al., 2025; Zhou et al., 2025; Shi et al., 2025; Munje et al., 2025). RoboVQA (Sermanet et al., 2024) evaluates multimodal long-horizon reasoning by utilizing a vast dataset of robot manipulation videos paired with textual descriptions. ERQA (Gemini Robotics Team et al., 2025) employs a multiple-choice format to assess the model's reasoning. ScanQA (Azuma et al., 2022) and SQA3D (Ma et al., 2022) extend evaluation into 3D environments. However, most of these efforts rely on offline VQA evaluation paradigms. These methods lack environmental feedback, preventing models from refining decisions based on action consequences. Furthermore, they often overlook the impact of physical constraints on execution. Ultimately, offline tasks disrupt the perception-action-reflection loop, failing to evaluate closed-loop control capabilities within dynamic environments.

Several works have attempted to integrate VLMs with interactive simulation environments(Zhang et al., 2025a; Zheng et al., 2022; Lu et al., 2025). LIBERO (Liu et al., 2023) and Calvin (Mees et al., 2022) enable agent-environment interaction via fine-grained proprioceptive actions. However, their heavy reliance on low-level control often induces severe cumulative errors, obscuring the potential of VLMs in high-level logical planning. Conversely, works such as Embodied-Reasoner (Zhang et al., 2025b) focus on high-level task planning, facilitating interaction through coarse-grained primitive actions, such as "Pickup {object}", by outputting target object names. In this paradigm, VLMs may attempt to "shortcut" the process, blindly outputting object names derived solely from semantic priors while neglecting visual grounding. Moreover, these approaches lack targeted feedback mechanisms, hindering the alignment of VLMs with simulation characteristics and limiting their ability to learn from error correction.

Building upon the AI2-THOR (Kolve et al., 2017) simulation environment, which features rich indoor scenes and atomic operations, we design an E2E online evaluation pipeline. We selected the BBox as the interaction modality, as it is widely adopted by current mainstream VLMs (Bai et al., 2025b; Hao et al., 2025; Guo et al., 2025; Bai et al., 2025a).

## 2.2. Evaluation Towards Real-World Applications

Several works (Shridhar et al., 2020; Deitke et al., 2022; Li et al., 2025) rely on instructions that are overly explicit and direct. Such straightforward instructions typically correspond to a unique solution, where a mere restatement of the instruction often suffices as the entire task plan. This simplicity creates a significant disparity with real-world applications (Li et al., 2024a).

In contrast, high-level semantic instructions align more closely with daily life and genuinely challenge the task decomposition capabilities of VLMs. However, within simulation environments rich in object diversity like AI2-THOR(Kolve et al., 2017), incorporating such instructions necessitates a benchmark capable of accommodating, parsing, and scoring multiple valid solutions. Although recent works (Majumdar et al., 2024; Kant et al., 2022) have attempted to address multi-solution evaluation within Open-ended QA or specific task domains, there remains a lack of a unified framework capable of supporting diverse general tasks while systematically evaluating complex, multi-solution execution trajectories.

We address this challenge through HTP. By designing universal directed acyclic graph (DAG) based task skeletons and projecting them onto specific scenes, we abstract task completion as the topological expansion of a DAG, thereby establishing a scoring scheme compatible with multiple valid solution paths.

## 2.3. Process Evaluation

Existing embodied AI evaluations predominantly rely on outcome-oriented metrics, such as task success rate, while process evaluations remain scarce and limited in scope (Li et al., 2024b). Process evaluation offers direct insights into model performance across critical capabilities, including perception and planning, thereby facilitating the precise diagnosis of bottlenecks encountered during long-horizon task execution. However, current mainstream interactive benchmarks (Shridhar et al., 2020; Li et al., 2024a) exclusively adopt E2E evaluation paradigms, lacking the evaluation for fine-grained diagnosis of intermediate execution stages.

QA paradigms, being offline methodologies, are inherently well-suited for process evaluation. Leveraging data derived from HTP and samples extracted from online interaction trajectories, we construct a QA-based process evaluation framework. Aligning with the operational logic of VLMs in embodied tasks, this framework encompasses five distinct capabilities: task decomposition, progress management, action prediction, perception grounding, and feedback correction.

## 3. The RealLife Benchmark

RealLife-Bench focuses on indoor household scenarios, requiring the agent to act as a personal life assistant, completing tasks based on text instructions through continuous interaction with the simulation environment. We collected common tasks from real household scenarios and, by integrating them with objects available in the simulation, constructed 1,021 detailed life-oriented tasks categorized into 15 types. Considering that the same task can have varied descriptions in real-world applications, we designed instructions across three levels of understanding difficulty for each task, as shown in Figure 1. For high-level semantic instructions such as "I am hungry," the solution space is diverse: the agent could provide ready-to-eat food, toast bread, or heat up other food items. Consequently, data that closely mirrors real life must support multiple target objects and multiple solutions. RealLife-Bench incorporates 17 primitive operations, including *pick up*, *put in*, *open*, *close*, *toggle on/off*, *slice*, *break*, *pour*, *move ahead/back/left/right*, *turn right/left/back*, *look up/down*, *navigate to* and *end*. In total, RealLife-Bench comprises 1,021 E2E online evaluation tasks and approximately 10,302 QA-based process evaluation samples.

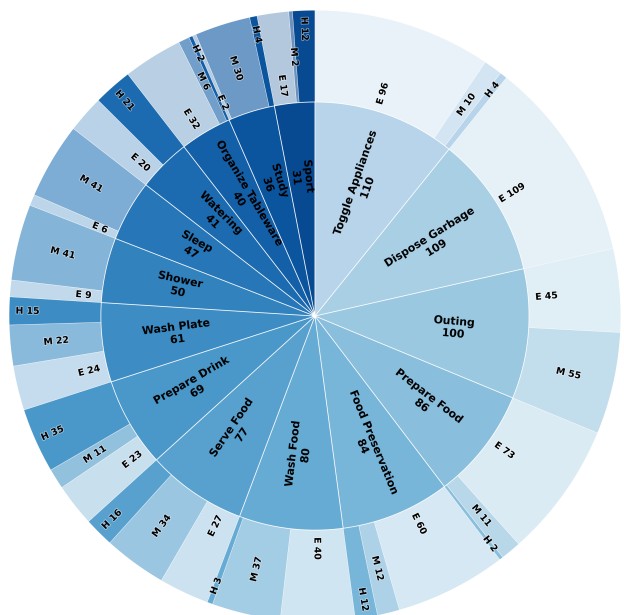

*Figure 1.* Distribution of task types and instruction difficulty levels in RealLife-Bench. The inner ring displays the names and total data counts for each task category, while the outer ring illustrates the specific sample distribution across three difficulty levels: Easy (E), Medium (M), and Hard (H).

## 3.1. Data Generation

**Hierarchical Topological Projection:** Guiding VLMs to complete scene-level embodied tasks requires high integration of instructions, scenes, and trajectories. We use HTP to generate task data in two layers: from abstract to concrete, and from logical to physical. As illustrated in Figure 2.

In Layer 1, we manually designed 64 distinct instruction templates, excluding trivial variations based solely on object substitution. Integrating these with the object ontology in AI2-THOR, we annotated the comprehensive completion logic for each instruction, structured as a DAG. Within this topological structure, each node represents a critical intermediate state required for task progression. We define $object to represent categories of objects rather than specific instances. Each node is governed by a set of state constraints (e.g., attributes like *isToggled* or relationships like *parentReceptacles*); an object instance is considered valid only if its attributes in the simulator align with these constraints. To support complex task logic, we implemented distinct node types for logical quantification: ANY dictates that the node expands if at least one valid object instance satisfies the condition, whereas ALL requires satisfaction by every applicable instance, among other logical constructs. Furthermore, these conditions support recursive nesting, enabling the construction of intricate, realistic task definitions that are grounded in common sense and independent of specific scene layouts. Subsequently, we identify which scenes are compatible with these DAGs. We traverse the initial configuration of all available scenes, matching the abstract requirements defined in the DAG against the concrete objects present. This process effectively "projects" the scene-agnostic task skeletons onto specific environments. Since a single DAG can encode multiple potential solution paths, a task is considered valid only if the target scene contains sufficient object instances to satisfy all abstract object variables along at least one complete solution path. This projection mechanism allows generic task skeletons to adaptively instantiate into executable, specific tasks across varying spatial layouts. Through this first layer of projection, we successfully fused instructions with scenes to generate 1,021 data.

Leveraging the node transition relationships annotated in the DAGs, in Layer 2, we sample multiple feasible solution paths from the data generated in Layer 1 and employ an automated planner to attempt execution. In the event of failure, the planner iteratively substitutes the target object with other instances from the same category until the task succeeds or all options are exhausted. For instance, if an interaction with an apple is required but the apple is currently out of view, the planner utilizes oracle metadata (this omniscient view is strictly for data generation and unavailable during evaluation) to navigate directly to the apple. By querying

the global environment state, if the planner detects that the apple is enclosed within a container, it automatically opens the container to establish a line of sight, executes the target action, and subsequently closes the container to restore the environment to its previous state. If the target object remains inaccessible, the planner backtracks to select an alternative object instance or re-initializes with a different solution path. Building upon the instruction-scene pairs from Layer 1, this second layer of projection generates 962 successful execution trajectories.

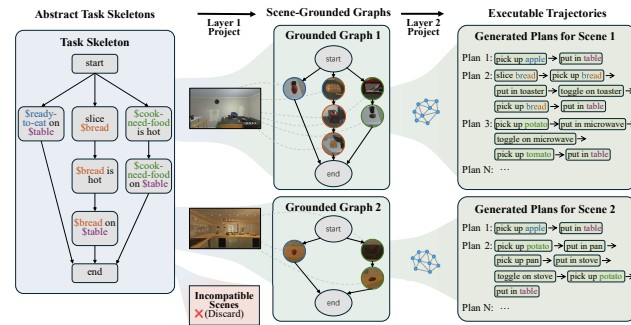

*Figure 2.* Overview of HTP. HTP comprises two phases: first, projecting manually annotated task skeletons onto specific scenes to generate instruction-scene pairs; second, deriving valid trajectories containing explicit plans via automated execution rollouts. In this figure, the depicted plan highlights only key steps for clarity.

In E2E online evaluation, the VLM acts as the brain of the agent in the simulation. Based on language instructions, current observations, and historical operation records, it generates the next step until task completion. Let $VLM(\cdot)$ represent the model. Each dialogue turn is defined as:

$$A_{t+1} = VLM(S, I, H_t)$$

Where $H_t = (O_0, A_1, F_1, O_1, ..., A_t, F_t, O_t)$. $t$ represents the turn, $A$ represents the primitive action and target object, $S$ and $I$ represent the system prompt and instruction respectively. $H_t$ is the history containing feedback $F$ and observation $O$.

RealLife-Bench requires the VLM to output the selected primitive action and the target object, represented by its BBox in the current observation image. After the VLM output, the pipeline parses the result, calculates intersection over union (IoU) based on the semantic segmentation map of the current observation to capture the target object, and the simulator executes the action and provides feedback. In previous works(Zhang et al., 2025b), failure feedback was vague, such as " Object: item is currently unavailable for interaction". However, if the failure stems from opening an already open object, the model fails to diagnose the specific cause of the error and may persist in futile attempts. In

this work, we collected 27 error types 12. The pipeline parses these errors and constructs direct cause feedback. If successful, a summary statement is added to the history as feedback 13.

Leveraging the DAG structure, we assess task completion via DAG expansion. After each action, we check the environment state to determine if the DAG can be expanded. The model's completion rate is the maximum expansion degree among solution paths. When any solution path is fully expanded, the model receives full marks.

To enrich scenes and simulate physical phenomena, we designed decoration actions. During initialization, objects are rearranged, appliances toggled, or objects dirtied to suit specific tasks. During tasks, we simulate cleaning effects: if an object is under a faucet and the faucet is on, the object becomes clean. We checked all full-score tasks to ensure tasks were not completed in the initial state.

### 3.2. Process Evaluation

We identify five key capabilities required for VLMs to complete embodied tasks, as illustrated in Figure 3.

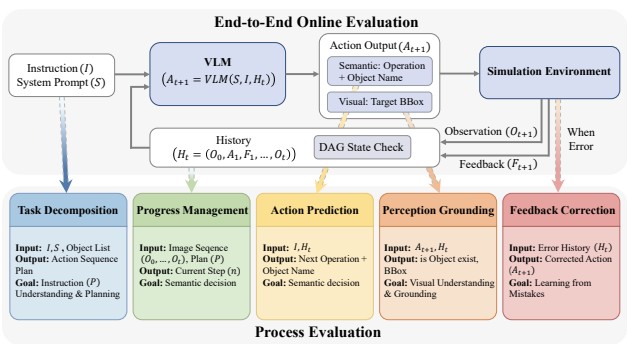

*Figure 3.* Overview of the relationship between process evaluation and E2E online evaluation. The E2E online evaluation operates as a closed loop whereas process evaluations are conducted for critical junctures within this cycle.

**Task Decomposition Evaluation (TD):** Upon receiving an instruction, the initial step involves request analysis and plan formulation. This evaluation targets the VLM's capability to decompose abstract instructions into actionable plans. Formally, each interaction is defined as:

$$P = VLM(S, I, Obj)$$

Where $P = (A_1, A_2, ...)$ denotes the generated action sequence, and $Obj$ represents the list of all object names within the scene. Including object names in the prompt serves as a textual description of the scene, requiring the

model to perform joint understanding of both the instruction and the environmental context.

RealLife-Bench requires the VLM to output an action sequence where target objects are identified by their names. This differs from the E2E evaluation as it decouples task decomposition from perception capabilities, specifically isolating instruction understanding from visual grounding.

The trajectory network generated by the HTP allows us to score based on the coverage of its output sequence against this network. We traverse the generated action sequence, comparing it with the network to validate node expansions. the final score is determined by the degree of node expansion achieved on the solution paths. In total, we evaluated 975 samples for TD.

**Progress Management Evaluation (PM):** Following plan formulation, the VLM must accurately monitor execution status. This evaluation examines the model's capacity to align long-sequence visual history with the intended trajectory to track progress. Formally, each interaction is defined as:

$$n = VLM(S, P, H_t)$$

Where $P = (A_1, A_2, ..., A_k, ...)$ represents the planned action sequence for the task, and $H_t = (O_0, O_1, ..., O_t)$ denotes the history. To prevent textual cues regarding actions or objects from leaking the current state's position within the plan, $H_t$ consists exclusively of image sequences, omitting textual feedback or action descriptions.

RealLife-Bench requires the VLM to identify the specific step index within the plan that corresponds to the current history. In the prompt, actions in the plan are sequentially numbered.

The dataset is derived from the trajectory data generated in Layer 2 of HTP. We sample specifically at timestamps $t$ corresponding to interaction actions (excluding navigation). To simulate scenarios involving planning deviations, actions in the provided plan $P$ subsequent to the correct index are either replaced with, or the plan is extended using, random, non-repetitive actions.

We employ a binary scoring metric where the model receives a score of 1 if it outputs the correct index $n$, and 0 otherwise. In total, we evaluated 1,216 PM samples.

**Action Prediction Evaluation (AP):** Upon accurately determining the current state, the VLM must identify the next executable operation. This evaluation assesses the model's ability to predict the correct next action based on instructions and historical context. Formally, each interaction is defined as:

$$A_{t+1} = VLM(S, I, H_t)$$

Where $H_t = (O_0, A_1, F_1, O_1, ..., A_t, F_t, O_t)$. To mitigate potential answer leakage where the textual description of $A_t$

in the history might explicitly suggest the subsequent target object, the textual content within the history is carefully managed.

RealLife-Bench requires the VLM to output $A_{t+1}$ as a combination of an primitive operation and the target object name. This format serves to decouple action prediction from perception capabilities, ensuring the evaluation does not conflate logical planning with visual grounding performance.

The dataset used for AP is consistent with that of PM. However, unlike PM, the explicit plan is excluded from the prompt, as its inclusion would directly reveal future actions.

For scoring, to address the issue where the model's naming convention for an object may differ from the ground truth (synonymy), we employ Qwen3 to align and score the results. Due to the absence of a global plan in the prompt, branching nodes in the task topology may yield multiple valid next actions. To alleviate the semantic burden on the judging model, we evaluate each valid solution independently using Qwen3. The scoring is binary, and the final score is derived via a logical OR operation across all valid solutions. In total, we evaluated 2,911 AP samples.

**Perception Grounding Evaluation (PG):** Upon deriving correct action semantics, the subsequent step requires grounding the action within the physical scene. This evaluation targets the VLM's capacity to spatially locate target objects within long, text-image interleaved sequences. Formally, each interaction is defined as:

$$exist, bbox = VLM(S, A_{t+1}, H_t)$$

Where $exist$ denotes a boolean indicating the presence of the target object specified in $A_{t+1}$. If $exist$ is true, the model must output the corresponding BBox. The history $H_t$ remains consistent with the online evaluation configuration. Distinct from traditional single-image grounding tasks, this evaluation challenges the model to filter out interference from historical contexts, specifically requiring the VLM to ground objects solely within the most recent observation rather than retrieving visual features from past frames.

The dataset for PG is derived from trajectories generated during the E2E online evaluation. We extract interaction histories from each model's episodes and select a target object present in the final observation frame. The ground truth BBox is generated utilizing the simulation environment's semantic segmentation maps. To simulate scenarios involving erroneous action predictions, we introduce negative samples with a probability of 0.3, where the query targets an object absent from the current observation.

For scoring, positive samples require the model to correctly predict $exist$ as True; the final score is then calculated as the IoU between the predicted and ground truth BBox. For negative samples, the model earns a score of 1 solely by

*Table 1.* Performance comparison across different tasks.

| MODEL | ON | TD | PM | AP | PG | FC |
|---|---|---|---|---|---|---|
| QWEN2.5 | 19.2 | 90.6 | 60.6 | 54.2 | 46.7 | 65.2 |
| MIMO | 22.1 | 82.7 | 62.7 | 58.9 | 42.8 | 47.3 |
| QWEN3 | 35.6 | 91.1 | 80.2 | 57.0 | 51.3 | 58.5 |
| SEED | 38.5 | 89.2 | **87.5** | 51.3 | 68.6 | 64.5 |
| GEMINI3F | **52.6** | **93.4** | 81.3 | **66.8** | **77.2** | **83.0** |

correctly predicting $exist$ as False; otherwise, the score is 0. In total, we evaluated approximately 2,600 PG samples.

**Feedback Correction Evaluation (FC):** During execution, agents inevitably encounter failures requiring rectification. This evaluation examines the VLM's capacity to incorporate environmental feedback into modified decision-making. The dialogue format remains consistent with the online evaluation.

The data for FC is derived from online evaluation trajectories. We sample specific timestamps $t$ where execution errors occurred, requiring the VLM to generate a revised action based on the error feedback. To ensure balanced evaluation, an equal number of error instances were sampled from the trajectories of each VLM.

RealLife-Bench employs Qwen3 as an judge model. We manually summarized common error types encountered during online evaluations and integrated them into the judge's prompt. The judge is instructed to first classify the error type and then assign a 0 or 1 score based exclusively on whether the target model's new decision demonstrates that it has heeded the feedback lesson. The evaluation process is formalized as:

$$score = \text{Qwen3}(S, H_t, A_{t+1})$$

To alleviate the semantic understanding load on the judge model, the history $H_t = (A_0, A_1, ..., A_t)$ is simplified to contain only the textual descriptions of past actions, resulting in a strictly text-based prompt. In total, we evaluated approximately 2,600 FC samples.

## 4. Experiments

**Main Results:** We evaluated five mainstream VLMs: Qwen2.5VL-72B(Bai et al., 2025b), MimoVL Embodied-7B(Hao et al., 2025), Qwen3VL-235B22A(Bai et al., 2025a), Seed1.5 Thinking Vision Pro(Guo et al., 2025), and Gemini 3.0 Flash. Prompts were designed to match official documentation keywords, requiring JSON output 4. The E2E online evaluation runs on AI2-THOR. Episodes terminate after 6 consecutive turns, 12 consecutive errors, or 30 total turns. Process evaluations are offline QA. Main results are in Table 1.

RealLife-Bench sets two metrics for task difficulty. First is

instruction difficulty, defined by the clarity of the target object and intention. Levels are Easy (both explicit), Medium (one explicit), and Hard (both implicit). Results for ON and TD across difficulties are in Table 2. Second is key object overlap rate, calculating the ratio of actions involving key objects to all object interactions, measuring the difficulty of locating and identifying key objects. Levels are Easy ($> 0.6$), Medium ($> 0.2$ and $< 0.6$), and Hard ($< 0.2$). Its relation to ON success is shown in Table 2.

We evaluated the results for positive and negative samples in PG, considering only the accuracy of object existence prediction and excluding the IoU calculation for positive samples. The results are shown in Figure 4.

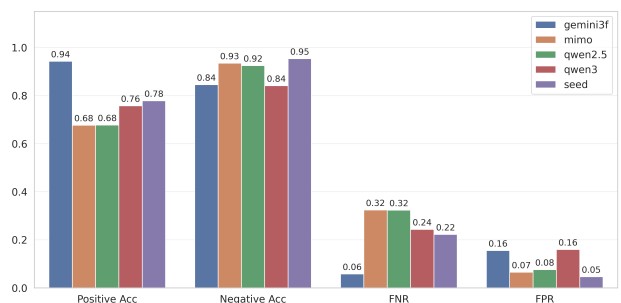

*Figure 4.* Detailed metrics for object existence prediction in Perception Grounding. FNR represents the proportion of existing objects incorrectly classified as absent by VLMs, while FPR denotes the proportion of non-existent objects incorrectly classified as present.

**Analysis:** Experimental data shows Gemini 3.0 Flash has a significant advantage in comprehensive performance, scoring highest in most dimensions , and achieving 52.6% in ON, far exceeding other models. Seed 1.5 follows, with the highest PM score 87.5%, showing strong status monitoring. Notably, while all models perform well in text-heavy planning dimensions like TD, all $> 82\%$, the gap widens significantly in dimensions involving dynamic perception and execution, PG and AP.

Gemini 3.0 Flash exhibits strong semantic parsing capabilities; even with Hard instructions, its TD score remains high at 82.4%, dropping only about 14.1% from Easy. In contrast, Qwen2.5 and Seed 1.5 drop precipitously in Hard mode (to 39.9% and 35.3% respectively), despite scoring over 93% in Easy mode. This indicates that some models have planning hallucinations or misunderstanding when facing implicit instructions. This demonstrates that introducing high-level semantic instructions is crucial for accurately assessing the planning capabilities of VLMs. Additionally, the drop in TD is generally smaller than in ON. For instance, Qwen2.5 retains 56.3% TD in Hard mode but only 7.9% ON success.

This confirms that even with a correct text plan, transforming it into a correct action flow during dynamic interaction remains a bottleneck.

Regarding the key object overlap rate, experimental results validate its effectiveness as a crucial metric for gauging the intrinsic difficulty of embodied tasks. As shown in Table 2, there is a uniform and drastic decline in success rates across all models as the task transitions from easy to hard. For instance, even the top-performing model, Gemini 3.0 Flash, experiences a precipitous drop from 90.4% in Easy mode to 22.8% in Hard mode. Other models suffer even more severe degradation. Unlike instruction difficulty, which primarily challenges semantic understanding, the overlap rate directly stresses the model's visual retrieval capabilities. The low success rates in the hard category underscore that locating and identifying objects within complex environments remains a significant block for current VLMs.

According to Figure 4, with the exception of Gemini 3.0 Flash, most models exhibit an imbalanced performance distribution. They achieve extremely high accuracy on negative samples, consistently exceeding 84% with a very low FPR of 4%-16%. However, their accuracy on positive samples hovers between 68% and 78%, resulting in a high FNR of 22%–32%. Given that objects with areas that are too small or too large were removed from the PG data, and Gemini 3.0 Flash's high accuracy rules out the possibility that the tasks themselves are inherently too difficult, this suggests that current VLMs tend to be conservative in embodied perception. This conservativeness is a significant factor contributing to the underperformance in PG, which constrains the completion of embodied tasks.

**Correlation Analysis:** To explore the relationship between process metrics and E2E success, we used Spearman ($\rho$) and Pearson ($r$)rank correlation. Results show PG has a significant perfect correlation with ON ($\rho = 0.9, r = 0.94$), revealing that precise visual perception is the prerequisite for task success. PM and AP follow ($\rho = 0.9, r = 0.83$ and $\rho = 0.9, r = 0.86$), forming the logical backbone of execution. TD has lower correlation ($\rho = 0.5, r = 0.63$), reflecting the planning-execution gap. Notably, we observe a trend where the correlation with task success strengthens as the process moves from abstract planning to concrete execution. This implies that while high-level semantic reasoning provides a foundational baseline, the terminal capability of visual perception acts as the critical bottleneck, ultimately determining the upper bound of performance in real-world applications.

FC also shows low correlation ($\rho = 0.4, r = 0.75$). This is partly attributed to the dependence of error rectification on underlying perception capabilities as corrected decisions may fail due to poor visual grounding even when the model demonstrates an intent to learn. Furthermore this trend sug-

*Table 2.* Comprehensive performance comparison across instruction difficulties and overlap rates.

| | INSTRUCTION SUCCESS (%) | | | TD SUCCESS (%) | | | OVERLAP SUCCESS (%) | | |
| MODEL | EASY | MED | HARD | EASY | MED | HARD | EASY | MED | HARD |
|---|---|---|---|---|---|---|---|---|---|
| GEMINI3F | **54.9** | **55.0** | **36.5** | **96.5** | 90.7 | **82.4** | **90.4** | **56.9** | **22.8** |
| MIMO | 26.4 | 19.0 | 10.6 | 85.8 | 78.4 | 75.9 | 66.2 | 13.2 | 5.3 |
| QWEN2.5 | 21.5 | 19.3 | 7.9 | 96.2 | **91.6** | 56.3 | 54.5 | 11.6 | 6.1 |
| QWEN3 | 40.4 | 32.6 | 20.9 | 96.4 | 88.2 | 68.2 | 76.6 | 34.7 | 9.0 |
| SEED | 43.8 | 33.6 | 26.2 | 93.7 | 91.5 | 58.3 | 81.0 | 37.9 | 11.7 |

gests a utility deviation where corrected decisions often fail to effectively advance the expansion of task nodes despite showing adaptive intent, rendering their actual contribution difficult to quantify within the current metric system.

**Case Study**: Taking fruit washing as an example, we compared task completion under identical conditions but different instructions. Significant gaps exist between models: Qwen3 successfully completes the task with simple, direct instructions but fails under hard instructions, even exhibiting severe hallucinations. In contrast, Gemini 3.0 Flash, with its stronger overall capabilities, successfully completes tasks in both scenarios, as shown in Figure 5. This demonstrates that high-level semantic instructions, which closer resemble real-life expressions, effectively distinguish the capabilities of different models.

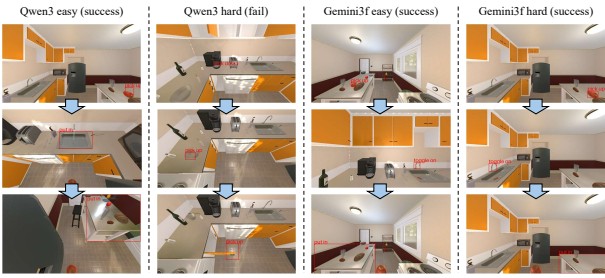

*Figure 5.* Impact of different instructions. **Easy instruction:** Wash a fruit and put it on the dining table." **Hard instruction:** I am thirsty and want to eat something clean and juicy, please prepare it and put it on the dining table." The DAGs for both instructions are identical.

In some cases where VLM decision execution fails, precise feedback can directly assist the model in rectifying errors and swiftly returning to the correct path for task completion. Furthermore, this mechanism enhances the VLM's environmental awareness and understanding of its capability scope, as illustrated in Figure 6.

## 5. Conclusion

In this work, we introduced **RealLife-Bench**, a holistic evaluation framework designed to assess VLMs in open-ended

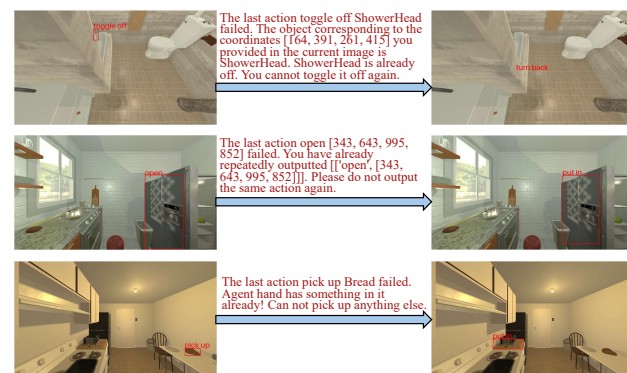

*Figure 6.* Assistance of precise feedback. **Top:** The VLM attempts to turn off the bathtub faucet, but it is already off. **Middle:** The VLM intends to put an apple in the fridge but mistakenly believes the fridge was not successfully opened, consecutively outputting the exact same "open fridge" decision twice. **Bottom:** The VLM attempts to pick up sliced bread but is already holding a knife, but the simulation environment allows holding at most one object.

household tasks. By utilizing HTP, we project annotated instructions onto diverse simulation environments to generate data and executable trajectories. To align with real-world applications, we abstract task completion as the topological expansion of DAG to support multi-solution tasks, and we also incorporate high-level semantic instructions and a precise feedback mechanism.

Our process evaluation diagnoses specific capability bottlenecks in embodied task execution. Experiments reveal a high correlation between specific capability dimensions and E2E online evaluation results. Notably, PG acts as the most critical capability of E2E success, serving as the terminal capability that strictly constrains performance despite the strong abstract decomposition skills of current VLMs.

These findings highlight the necessity of enhancing the perception-action-reflection loop in future research. We hope RealLife-Bench serves as a valuable testbed for the community, facilitating the development of VLMs.

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

## A. Prompt and demo

**End-to-End Online Evaluation**:

---

### End-to-End Online Evaluation Prompt

You are an indoor life assistant. You need to select the next action from the provided primitive operations based on your historical execution records and the current latest observed image. Sometimes the user's instruction is a first-person narrative; you need to understand the user's needs from the instruction, complete the task step-by-step, and satisfy the user's requirements. This is your task: Help me prepare breakfast with both food and drink, and put it on the dining table

Your history action, corresponding feedback and observation is as follows: This is the image at your initial position

{image 0}

Your response: {"thought": "The user wants a breakfast with food and a drink placed on the dining table. I can see an apple on the counter which can serve as food. I need to find a drink as well. I'll start by picking up the apple.", "primitive action": "pick up", "target object": [756, 473, 811, 501]} feedback: The object corresponding to the coordinates you provided in the current image is Apple. Your last action pick up Apple was successful, you have picked up the Apple. You received the following new observation image.

{image 1}

...

{image last}

Please select a suitable primitive operation based on the historical records and the latest observation image. You can choose the most appropriate one from the following primitive operations. The system will select the object with the highest IoU (Intersection over Union) with your provided bounding box as the target object or target receptacle. **If you see an object, you can directly output a suitable bbox to interact with it; you do not need to be close to the object.** "pick up" When your hand is empty, pick up the object within the **specified bounding box** and hold it in your hand. Target object's **bounding box** needed."put in" When you are holding an object, place the object in your hand into the container/location within the **specified bounding box**. Target container/location's **bounding box** needed. "open" Open the object within the **specified bounding box**. Applicable to things such as cabinet, window, curtain. Target object's **bounding box** needed. "close" Close the object within the **specified bounding box**. Applicable to containers that are already open, such as cabinet, window, curtain. Target object's **bounding box** needed. "toggle on" Turn on the object within the **specified bounding box**. Applicable to home appliances and faucets. Target object's **bounding box** needed. "toggle off" Turn off the object within the **specified bounding box**. Applicable to home appliances and faucets. Target object's **bounding box** needed. "slice" Slice the object within the **specified bounding box**. Slice the object into pieces. Target object's **bounding box** needed. "break" Break the object within the **specified bounding box**. Break glass or ceramic objects, or crack eggs. Target object's **bounding box** needed. "pour" Pour water from the held object nearby, such as watering plants. Target object prohibited. "move ahead" Move forward a fixed distance without rotating the observation view. Target object prohibited. "turn right" Rotate right in place by 90 degrees, changing the observation view without changing position. Target object prohibited. "turn left" Rotate left in place by 90 degrees, changing the observation view without changing position. Target object prohibited. "turn back" Trun back in place by 180 degrees, changing the observation view without changing position. Target object prohibited. "look up" Tilt the view upward in place by 30 degrees, changing the observation view without changing position. Target object prohibited. "look down" Tilt the view downward in place by 30 degrees, changing the observation view without changing position. Target object prohibited. "navigate to" Move next to the object specified by the **bounding box** and orient the view towards that object. Target object's **bounding box** needed. "end" When you have completed the task, output end to finish the task. Target object prohibited. Your task can be completed within the current room. You cannot go to other rooms, but you can change your viewpoint or open furniture to search for the objects you need. Please summarize what you have done and what your plan is. Output your thought process, your chosen primitive action. Detect the target object in the image. The box 2d should be [ymin, xmin, ymax, xmax] normalized to 0-1000. All in JSON format. You can only output one primitive operation at a time. Please make a decision and identify the target object based on the latest observation below.

{image last}

---

*Table 3.* End-to-End Online Evaluation Prompt

**Prompts for Guiding VLM Grounding in End-to-End Online Evaluation**

gemini3f: Detect the target object in the image. The box_2d should be [ymin, xmin, ymax, xmax] normalized to 0-1000.
qwen2.5: Locate the target object, output its bbox coordinates using JSON format.
mimo: Output a JSON in the format [{{"action": [...], "bbox_2d": [...], "label": "{{the_whole_description}}"}}, ...].
seed: For the target object, provide its bounding box in the format: ¡bbox¿x1 y1 x2 y2¡/bbox¿.
qwen3: Detect the target object in the image and output their bbox coordinates in JSON format.

*Table 4.* Prompts for Guiding VLM Grounding in End-to-End Online Evaluation

**Tsak decomposition Evaluation Prompt**

You are a robot in a given room. You need to complete the tasks according to human instructions. We provide an Available Actions set and the corresponding explanations for each action. You need to provide the complete task plan based on the instructions and current observations, starting from the task's beginning. Focus strictly on the instruction's requirements; do not add unnecessary steps to "perfect" the plan. You only need to provide the core actions; finding objects is not required.
This is your task: Help me prepare breakfast with both food and drink, and put it on the dining table
The primitive operations you can choose from and their descriptions are as follows: You can select primitive operations from the following choices to compose an action sequence. The object placeholders need to be replaced with the names of interactable objects in the scene. Tips: If you want to fill water, please put the container in the sink and toggle on the faucet. If you want to get coffee, put the container in the coffee machine first, then toggle on the coffee machine., "pick up object" Pick up object and hold it in your hand, "put object 1 in object 2" Put object 1 on or inside object 2, "open object" Open the object. Applicable to containers that store other objects, "close object" Close the object. Applicable to containers that are already open, "toggle on object" Turn on the object. Applicable to home appliances and faucets, "toggle off object" Turn off the object. Applicable to home appliances and faucets, "slice object" Slice the object into pieces, such as bread and other food., "break object" Break glass or ceramic objects, or crack eggs., "pour" Pour water from the held object nearby, such as watering plants., "end" When you have completed the task, use end to finish the task. The objects in the scene are Apple, Bowl, Bread, ButterKnife, Cabinet, CoffeeMachine, CounterTop, Cup, DiningTable, DishSponge, Drawer, Egg, Faucet, Floor, Fork, Fridge, GarbageCan, Kettle, Knife, Lettuce, LightSwitch, Microwave, Mug, Pan, PaperTowelRoll, PepperShaker, Plate, Pot, Potato, SaltShaker, Sink, SoapBottle, Spatula, Spoon, Stool, StoveBurner, StoveKnob, Toaster, Tomato, Window. Please select from the primitives above to generate a sequence of actions to complete the task given by the user. Output the result in JSON format, where each action is represented by a string, and these actions form a list.

*Table 5.* Progress Management Evaluation Prompt

**Progress Management Evaluation Prompt**

You will be provided with a numbered action list and a sequence of first-person images showing an agent executing this action list sequentially in a room. Your task is to determine the progress by matching the visual evidence to the actions **in order from the beginning**.
Here are the first-person views of the agent executing the action sequence:
{image 0}
...
{image t}
Here is the action list: 1. pick up PaperTowelRoll 2. put in PaperTowelRoll 3. put in GarbageCan 4. slice RemoteControl. Please analyze the agent's completion status. Output the **maximum index of the strictly consecutive actions** that have been completed. Choose the index that fits the current chronological sequence of events. Please output the result in JSON format

*Table 6.* Progress Management Evaluation Prompt

**Instruction: The Potato is dirty, but I need it for cooking**

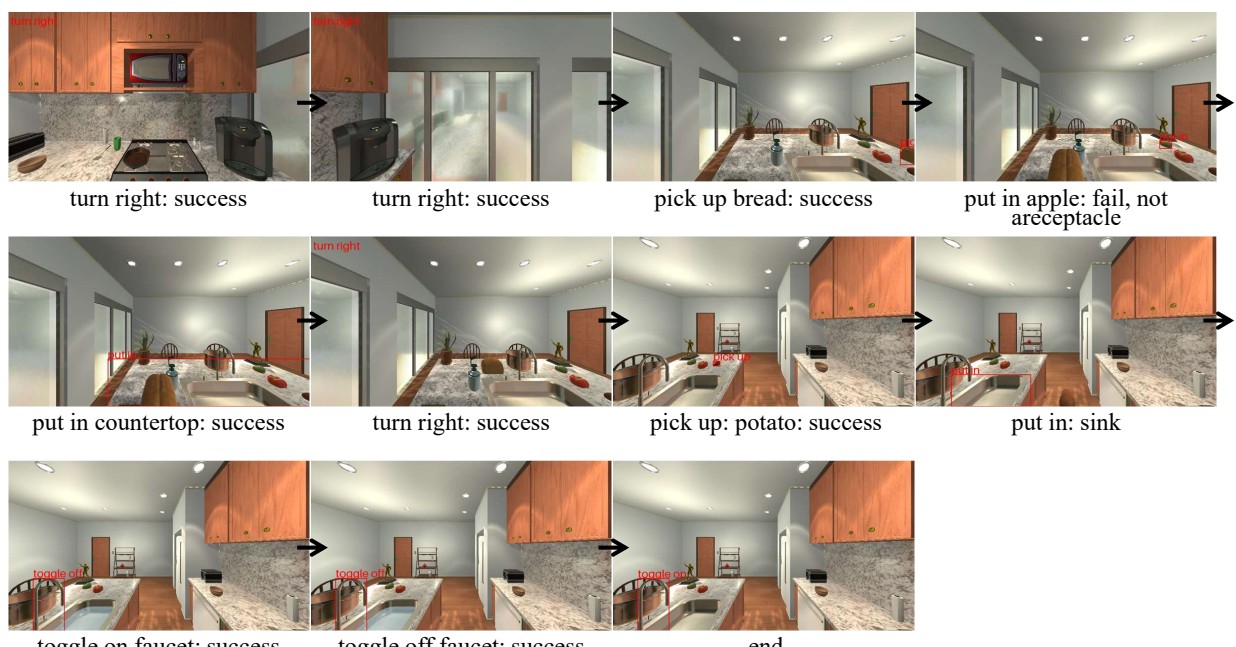

turn right: success      turn right: success      pick up bread: success      put in apple: fail, not areceptacle

put in countertop: success      turn right: success      pick up: potato: success      put in: sink

toggle on faucet: success      toggle off faucet: success      end

*Figure 7.* An example of E2E online evaluation

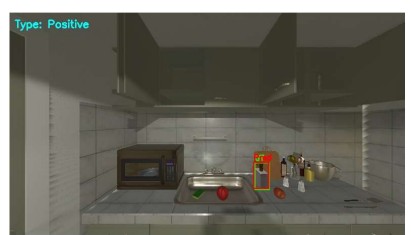 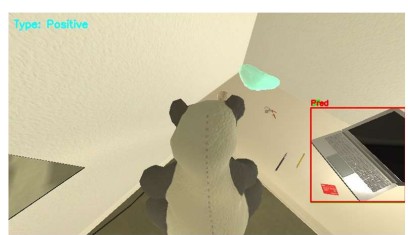 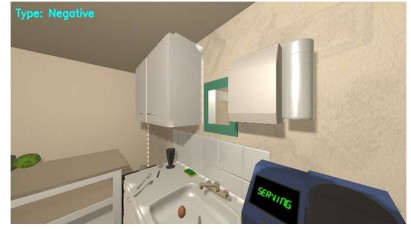

"score": 0.9079516226126001,
"target_object_id": "SoapBottle",
"model_response":
"```json\n{\"exist\": true, \"box_2d\":
[673, 617, 791, 654]}\n```",

"score": 0.9911175680491217,
"target_object_id": "Laptop
"model_response": "{\n \"exist\":
true,\n \"box_2d\": [\n 428,\n 762,\n
850,\n 1000\n ]\n}",

"score": 1.0,
"target_object_id": "DeskLamp",
"model_response": "{\"exist\": false}",

*Figure 8.* An example of perception grounding. The **green** box represents the ground truth BBox, while the **red** box denotes the VLM's prediction.

**Action Prediction Evaluation Prompt**

You are an indoor life assistant. You need to select the next action from the provided primitive operations based on your historical execution records and the current latest observed image. Sometimes the user's instruction is a first-person narrative; you need to understand the user's needs from the instruction, complete the task step-by-step, and satisfy the user's requirements.

This is your task: The potato is dirty, but I need it for cooking

Your history are as follow:

This is the image at your initial position

{image 0}

Your action: navigate

{image 1}

Your action: pick up Potato

{image 2}

Your action: navigate

{image 3}

Your action: put in Sink

{image 4}

Please select a suitable primitive operation based on the historical records and the latest observation image. You can select primitive operations from the following choices to compose an action sequence. The {object} placeholders need to be replaced with the names of interactable objects in the scene. "pick up {object}": Pick up object and hold it in your hand "put in {object}": When you are holding an object, put the thing in your hand on or inside object "open {object}": Open the {object}. Applicable to containers that store other objects, such as cabinets "close {object}": Close the {object}. Applicable to containers that are already open, such as cabinets "toggle on {object}": Turn on the {object}. Applicable to home appliances and faucets "toggle off {object}": Turn off the {object}. Applicable to home appliances and faucets "slice {object}": Slice the {object} into pieces, such as bread and other food. "break {object}": Break glass or ceramic objects, or crack eggs. "pour": Pour water from the held object nearby, such as watering plants. "end": When you have completed the task, use end to finish the task Your task can be completed within the current room. Please summarize what you have done. Output your thought process, your chosen primitive action and object. All in JSON format. You can only output one primitive operation at a time. Please make a decision and identify the target object based on the latest observation below.

{image 4}

*Table 7.* Action Prediction Evaluation Prompt

**Action Prediction Evaluation Metric Prompt**

You are a scoring model. You will receive the ground truth, along with the output to be evaluated. Please judge whether the output action and object are consistent with the truth. The ground truth action must be contained within the output action string (it is sufficient if the output action includes the ground truth action), while the object name should be judged based on semantic equivalence, as the same object may be referred to by different names. When the object in the answer is None, the object in the output can be anything. If both are correct, output 1; otherwise, output 0.

ground truth: ['action': 'toggle on', 'object': 'Faucet'], output: "thought": "The potato is in the sink. To wash it, I need to turn on the water. I will toggle on the faucet.", `primitive action": "toggle on faucet", "object": "faucet"

*Table 8.* Action Prediction Evaluation Metric Prompt

Perception Grounding Evaluation Prompt

You are an indoor life assistant. You are executing instructions given by the user.
Your history action, corresponding feedback and observation is as follows:
This is the image at your initial position
{image 0}
Your response: "thought": "I need to navigate to the bed to check what items are on it and ensure only the pillow and alarm clock remain.", "action": "navigate to", "bbox": [ 104, 937, 205, 997 ]
feedback: The object corresponding to the coordinates you provided in the current image is Bed. Your last action navigate to Bed was successful, you have navigated to the Bed. You received the following new observation image."
{image 1}
...
{image last}
Your current decision is: pick up Vase. Please detect the target object in the image. First, please use the JSON key "exist" to indicate whether the object is present. If the target object exists in the current scene, the box 2d should be [ymin, xmin, ymax, xmax] normalized to 0-1000. If the target object does not exist in the current scene, indicate that it is not found. Output everything in JSON format. Please identify the target object based on the latest observation below.
{image last}

*Table 9.* Perception Grounding Evaluation Prompt

> ### Feedback Correction Evaluation Prompt
>
> You are an indoor life assistant. You need to select the next action from the provided primitive operations based on your historical execution records and the current latest observed image. Sometimes the user's instruction is a first-person narrative; you need to understand the user's needs from the instruction, complete the task step-by-step, and satisfy the user's requirements.
> This is your task: I am going to take a bath soon, put the soap on the bathtub, and the bathtub is currently empty
> Your history action, corresponding feedback and observation is as follows:
> This is the image at your initial position
> {image 0}
> Your response: {"thought": "The user wants to put soap on the bathtub. I need to find the soap first. It's likely in the cabinet above the toilet. I'll start by opening that cabinet.", "primitive": "open", "element": [128, 110, 442, 365]}
> The last action open failed. The object corresponding to the coordinates [128, 110, 442, 365] you provided in the current image is CabinetBodyMesh. Due to limitations of the simulation environment, target object CabinetBodyMesh is not interactable, Please try another action or object. The following observation image has not changed.
> {image 1}
> Please select a suitable primitive operation based on the historical records and the latest observation image. You can choose the most appropriate one from the following primitive operations. The system will select the object with the highest IoU (Intersection over Union) with your provided bounding box as the target object or target receptacle. **If you see an object, you can directly output a suitable bbox to interact with it; you do not need to be close to the object.** "pick up": When your hand is empty, pick up the object within the **specified bounding box** and hold it in your hand. Target object's **bounding box** needed. "put in": When you are holding an object, place the object in your hand into the container/location within the **specified bounding box**. Target container/location's **bounding box** needed. "open": Open the object within the **specified bounding box**. Applicable to things such as cabinet, window, curtain. Target object's **bounding box** needed. "close": Close the object within the **specified bounding box**. Applicable to containers that are already open, such as cabinet, window, curtain. Target object's **bounding box** needed. "toggle on": Turn on the object within the **specified bounding box**. Applicable to home appliances and faucets. Target object's **bounding box** needed. "toggle off": Turn off the object within the **specified bounding box**. Applicable to home appliances and faucets. Target object's **bounding box** needed. "slice": Slice the object within the **specified bounding box**. Slice the object into pieces. Target object's **bounding box** needed. "break": Break the object within the **specified bounding box**. Break glass or ceramic objects, or crack eggs. Target object's **bounding box** needed. "pour": Pour water from the held object nearby, such as watering plants. Target object prohibited. "move ahead": Move forward a fixed distance without rotating the observation view. Target object prohibited. "turn right": Rotate right in place by 90 degrees, changing the observation view without changing position. Target object prohibited. "turn left": Rotate left in place by 90 degrees, changing the observation view without changing position. Target object prohibited. "turn back": Trun back in place by 180 degrees, changing the observation view without changing position. Target object prohibited. "look up": Tilt the view upward in place by 30 degrees, changing the observation view without changing position. Target object prohibited. "look down": Tilt the view downward in place by 30 degrees, changing the observation view without changing position. Target object prohibited. "Maps to": Move next to the object specified by the **bounding box** and orient the view towards that object. Target object's **bounding box** needed. "end": When you have completed the task, output end to finish the task. Target object prohibited. Your task can be completed within the current room. You cannot go to other rooms, but you can change your viewpoint or open furniture to search for the objects you need. Please summarize what you have done and what your plan is. Output your thought process, your chosen primitive action. Detect the target object in the image. The box2d should be [ymin, xmin, ymax, xmax] normalized to 0-1000. All in JSON format. You can only output one primitive operation at a time. Please make a decision and identify the target object based on the latest observation below.
> {image 1}

*Table 10.* Feedback Correction Evaluation Prompt

**Feedback Correction Evaluation Metric Prompt**

You are a scoring model. You need to judge whether the target model can learn from error information and correct its mistakes. Acting as a life assistant completing a task, the target model will provide you with the task instruction and its history of operations. The target model has just made an error in its decision-making; it will perform self-correction based on these errors and provide a new decision. You will receive this information and the interactable objects in the current scene to score the target model's self-correction.

Below are the target model's response when it made a mistake, as well as the error message

{"thought": "The user wants to put soap on the bathtub. I need to find the soap first. It's likely in the cabinet above the toilet. I'll start by opening that cabinet.", "primitive": "open", "element": [128, 110, 442, 365]}

The last action open failed. The object corresponding to the coordinates [128, 110, 442, 365] you provided in the current image is CabinetBodyMesh. Due to limitations of the simulation environment, target object CabinetBodyMesh is not interactable, Please try another action or object. The following observation image has not changed.

Below are the output of the target model's self-correction

{"thought": "I need to find the soap. Since opening the cabinet directly failed, I will look around the room to see if it's placed elsewhere. I'll start by turning right to see the rest of the bathroom.", "primitive": "turn right", "element": []}

Please judge based on the target model's self-correction output. You do not need to consider whether the new decision contributes to task completion; simply focus on whether the lesson has been learned. However, if the new decision repeats the previous action, it is likely incorrect. After parsering, the target model's decision is turn right. There are some typical error types: 1. Stuck in a loop: repeatedly outputting incorrect decisions or constantly adjusting the view without acting. 2. Perception: the object the target model wants to interact with is not in the scene at all (the model needs to adjust the view or open furniture to search). 3. Grounding: the object is in the scene but grounded incorrectly. 4. General errors: e.g., nowhere to place an object, object cannot be opened, etc. From an omniscient perspective, the objects currently in the scene are: Bathtub, Cabinet, Candle, Cloth, CounterTop, Faucet, Floor, GarbageCan, HandTowel, HandTowelHolder, LightSwitch, Mirror, Plunger, ScrubBrush, ShowerCurtain, Sink, SoapBar, SoapBottle, SprayBottle, Toilet, ToiletPaper, ToiletPaperHanger, Towel, TowelHolder. Please first analyze the target model's error type. Then, if the model successfully learned the lesson, output <score>1</score>; otherwise, output <score>0</score>. Ensure the output is strictly enclosed in XML tags. Please also provide your reason for the score."

*Table 11.* Feedback Correction Evaluation Metric Prompt

# B. Targeted Feedback

*Table 12.* Error Reasons and Transformed Feedback Messages

| Error Reason | Transformed Feedback Information |
| --- | --- |
| Interaction with Non-Interactable Structure (e.g., Walls) | The last action {action} failed. The object corresponding to the coordinates {coordinates} you provided in the current image is {object_id}. Due to limitations of the simulation environment, target object {object_id} is not interactable, Please try another action or object. The following observation image has not changed. |
| Target Object Cannot Be Picked Up | The last action {action} {object_id} failed. The object corresponding to the coordinates {coordinates} you provided in the current image is {object_id}. {object_name} can not be picked, please try other object or action. The following observation image has not changed. |
| Hand Already Full (Pickup Failed) | The last action {action} {object_id} failed. Agent hand has something in it already! Can not pick up anything else. The following observation image has not changed. |
| Invalid Placement Logic | The last action {action} {object_id} failed. The container/location corresponding to the coordinates {coordinates} you provided in the current image is {object_id}. Put {held_object} in/on {target_receptacle} is not logic in this simulator. The following observation image has not changed. |
| Target Receptacle is Closed | The last action {action} {object_id} failed. The container corresponding to the coordinates {coordinates} you provided in the current image is {object_id}. {object_id} is closed, can not place if target is not open. The following observation image has not changed. |
| Agent Not Holding Anything (Place Failed) | The last action {action} {object_id} failed. You are not holding any object, so you cannot place anything. The following observation image has not changed. |
| Placement Position Invalid (Full/View Blocked) | The last action {action} {object_id} failed. Cannot find a valid position to place the object. The target receptacle might be full, you need to replace the other objects in the receptacle, or you may need to get closer or adjust your view of the target. The following observation image has not changed. |
| Placement Blocked by Collision | The last action {action} {object_id} failed. Another object's collision is blocking held object from being placed, please try other actions or target position. The following observation image has not changed. |
| Target is Not a Receptacle | The last action {action} {object_id} failed. The target position corresponding to the coordinates {coordinates} you provided in the current image is {object_id}. {object_id} is not a receptacle, please try other actions or target position. The following observation image has not changed. |
| Object Not Toggleable | The last action {action} {object_id} failed. The object corresponding to the coordinates {coordinates} you provided in the current image is {object_id}. {object_id} cannot be toggled on or off. The following observation image has not changed. |

| Error Reason | Transformed Feedback Information |
| --- | --- |
| Object Already On | The last action {action} {object_id} failed. The object corresponding to the coordinates {coordinates} you provided in the current image is {object_id}. {object_id} is already on. You cannot toggle it on again. The following observation image has not changed. |
| Object Must Be Closed to Toggle On | The last action {action} {object_id} failed. The object corresponding to the coordinates {coordinates} you provided in the current image is {object_id}. {object_id} must be closed before it can be toggled on. The following observation image has not changed. |
| Object Already Off | The last action {action} {object_id} failed. The object corresponding to the coordinates {coordinates} you provided in the current image is {object_id}. {object_id} is already off. You cannot toggle it off again. The following observation image has not changed. |
| Object Not Openable | The last action {action} {object_id} failed. The object corresponding to the coordinates {coordinates} you provided in the current image is {object_id}. The object {object_id} cannot be opened or closed. The following observation image has not changed. |
| Object Must Be Off to Open | The last action {action} {object_id} failed. The object corresponding to the coordinates {coordinates} you provided in the current image is {object_id}. {object_id} must be turned off before it can be opened. The following observation image has not changed. |
| Object Cannot Be Sliced | The last action {action} {object_id} failed. The object corresponding to the coordinates {coordinates} you provided in the current image is {object_id}. The object {object_id} cannot be sliced. The following observation image has not changed. |
| Object Cannot Be Broken | The last action {action} {object_id} failed. The object corresponding to the coordinates {coordinates} you provided in the current image is {object_id}. The object {object_id} cannot be broken. The following observation image has not changed. |
| Movement Blocked / Out of Bounds | The agent has reached the end in this direction or blocked by something. You received the following new observation image. |
| Rotation Blocked by Held Item | The last action {action} failed. If rotating {degrees} degrees at the current position, the object {held_object_name} in the agent's hand will be blocked. The following observation image has not changed. |
| Invalid Navigation Target | The last action {action} {object_id} failed. The object corresponding to the coordinates {coordinates} you provided in the current image is {object_id}. Cannot navigate to {object_id}. It is not a valid navigation target. The following observation image has not changed. |
| Object Already Empty | The last action {action} {object_id} failed. The object corresponding to the coordinates {coordinates} you provided in the current image is {object_id}. Cannot navigate to {object_id}. {object_id} already empty. The following observation image has not changed. |
| Pouring With Empty Hand | The last action {action} failed. There is nothing held in your hand now. The following observation image has not changed. |
| Held Object Contains No Liquid | The last action {action} failed. The object you hold is not filled with any liquid. The following observation image has not changed. |

| Error Reason | Transformed Feedback Information |
|---|---|
| Parser Error (No Match) | No match action found in Your response. Please follow the output format The following observation image has not changed. |
| Parser Error (Invalid Primitive/Format) | The last action {action} {coordinates} failed. Action is not a valid primitive or the format of target object not match. Please follow the output format. The following observation image has not changed. |
| Action Repeated | The last action {action} {coordinates} failed. {action} {coordinates} is the same as the last action, and is not repeatitive. Please chose other action or change coordinate. |
| Invalid JSON Format | The response is not in valid JSON format. Please ensure it is enclosed in curly braces { }. |
| Decoration Failed | Decoration error: decoration failed |

*Table 13.* Successful Actions and Feedback Messages

| Action Type | Feedback Message |
|---|---|
| Pick Up | The object corresponding to the coordinates you provided in the current image is {object_id}. Your last action {action} {object_id} was successful, you have picked up the {object_id}. You received the following new observation image. |
| Put In | The object corresponding to the coordinates you provided in the current image is {object_id}. Your last action {action} {object_id} was successful, you have put the object you were holding in/on the {object_id}. You received the following new observation image. |
| Open | The object corresponding to the coordinates you provided in the current image is {object_id}. Your last action {action} {object_id} was successful, you have opened the {object_id}. You received the following new observation image. |
| Close | The object corresponding to the coordinates you provided in the current image is {object_id}. Your last action {action} {object_id} was successful, you have closed the {object_id}. You received the following new observation image. |
| Toggle On | The object corresponding to the coordinates you provided in the current image is {object_id}. Your last action {action} {object_id} was successful, you have toggled the {object_id} on. You received the following new observation image. |
| Toggle Off | The object corresponding to the coordinates you provided in the current image is {object_id}. Your last action {action} {object_id} was successful, you have toggled the {object_id} off. You received the following new observation image. |
| Slice | The object corresponding to the coordinates you provided in the current image is {object_id}. Your last action {action} {object_id} was successful, you have sliced the {object_id}. You received the following new observation image. |

| Action Type | Feedback Message |
|---|---|
| Break | The object corresponding to the coordinates you provided in the current image is {object_id}. Your last action {action} {object_id} was successful, you have broken the {object_id}. You received the following new observation image. |
| Pour | Your last action {action} was successful, you have poured out the water from the object in your hand. You received the following new observation image. |
| Move Ahead | Your last action {action} was successful, you have moved ahead a fixed distance. You received the following new observation image. |
| Move Back | Your last action {action} was successful, you have moved back a fixed distance. You received the following new observation image. |
| Move Left | Your last action {action} was successful, you have moved left a fixed distance. You received the following new observation image. |
| Move Right | Your last action {action} was successful, you have moved right a fixed distance. You received the following new observation image. |
| Turn Right | Your last action {action} was successful, you have turned right a fixed degree. You received the following new observation image. |
| Turn Left | Your last action {action} was successful, you have turned left a fixed degree. You received the following new observation image. |
| Look Up | Your last action {action} was successful, you have looked up a fixed degree. You received the following new observation image. |
| Look Down | Your last action {action} was successful, you have looked down a fixed degree. You received the following new observation image. |
| Navigate To | The object corresponding to the coordinates you provided in the current image is {object_id}. Your last action {action} {object_id} was successful, you have navigated to the {object_id}. You received the following new observation image. |
| End | (No feedback message provided) |
| Default Success | Your last action {action} {object_id} was successful. You received the following new observation image. |

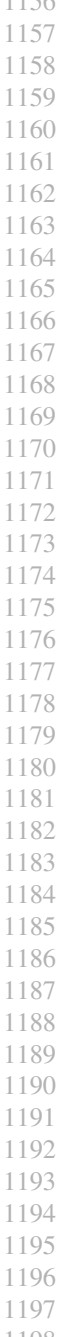
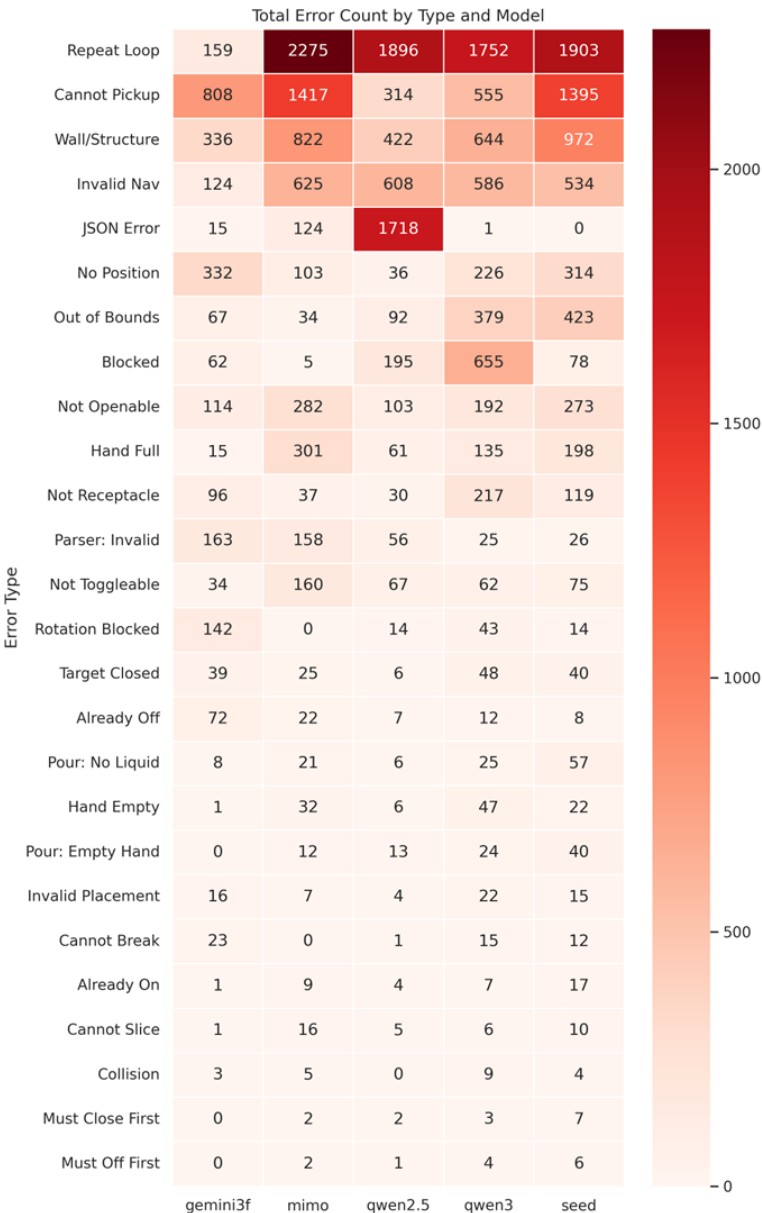

*Figure 9.* Statistics of total errors encountered by each model during the E2E online evaluation. The most prevalent error is the consecutive generation of identical incorrect outputs (repetition loops), a phenomenon characteristic of long-horizon tasks. Other error categories may stem from intrinsic logical flaws within the VLMs, perception grounding failures, or the interactive constraints of the simulation environment.

# C. Success Rates by Task Type

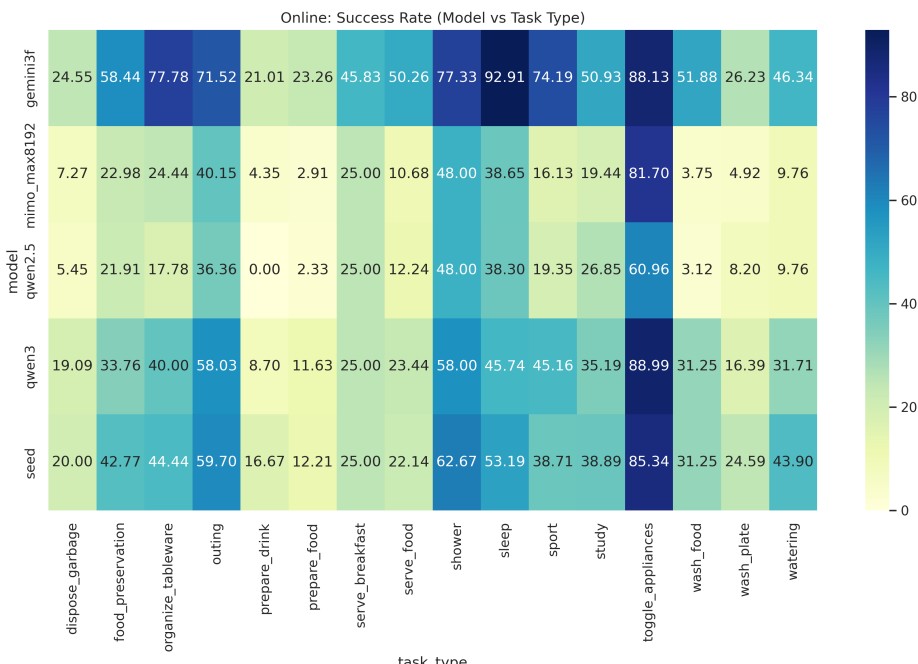

*Figure 10.* Success rates of VLMs across different task typies. While many models achieve high scores on tasks with simpler logic, significant performance gaps remain evident between models within the same category.

# D. Relationship between Success Rate and Sequence Length



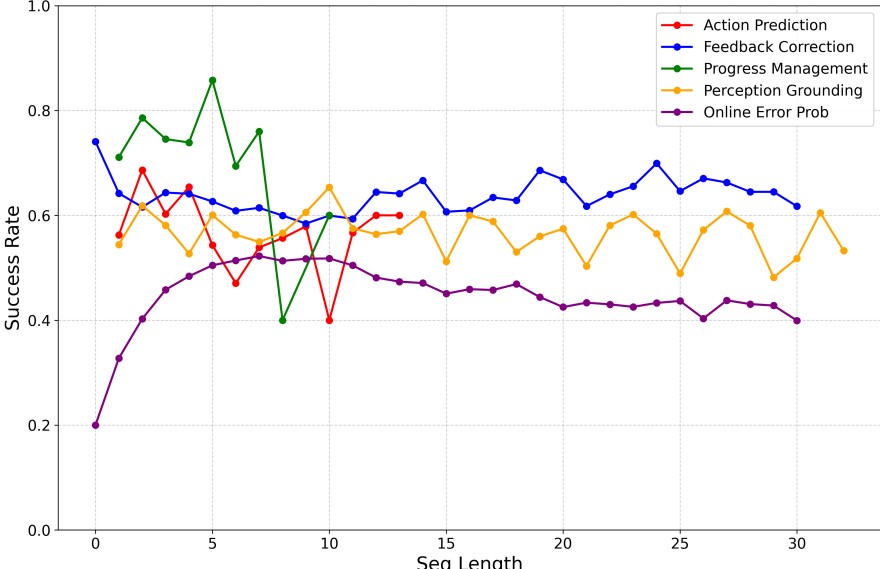

*Figure 11.* Success rates relative to sequence length across various tasks. This analysis includes results from all process evaluations (excluding TD) and the decision execution error rates from the E2E online evaluation. However, the results indicate no significant correlation between success rate and sequence length.

# E. Efficiency of Task Decomposition

*Table 14.* Efficiency comparison of TD across different models. Efficiency is defined as the ratio of actions aligned with the DAG to the total number of actions within the generated plan for successful tasks. To ensure a fair comparison, metrics were computed on a common subset of over 600 samples where all models achieved success. Results indicate negligible differences in efficiency across models; notably, Gemini 3.0 Flash does not exhibit the significant lead observed in other capability dimensions.

| MODEL | GEMINI3F | MIMO | QWEN2.5 | QWEN3 | SEED |
|---|---|---|---|---|---|
| EFFICIENCY | 60.4 | 57.2 | 63.4 | 61.8 | 63.1 |

