# OpenReview forum: "RealLife-Bench: A Benchmark for VLMs on Open-Ended Daily Activities with Holistic Evaluation"
_ICML.cc/2026/Conference — Submitted to ICML 2026_

### Official Review · Reviewer_qK6S · 2026-02-16

**Soundness:** 2
**Presentation:** 1
**Significance:** 3
**Originality:** 3
**Overall Recommendation:** 3
**Confidence:** 2

**Summary:**

This paper introduces RealLife-Bench, an evaluation framework for embodied AI that assesses VLMs across 1,021 online tasks covering 15 categories of open-ended daily activities in household settings. The research employs HTP technique to map abstract, high-level semantic instructions into DAG task structures, integrating a precise feedback mechanism to simulate a biological perception-action-reflection loop. Experimental analysis reveals that while current mainstream models demonstrate strong task decomposition capabilities.

**Compliance With Llm Reviewing Policy:**

Affirmed.

**Key Questions For Authors:**

See Weakness.

**Limitations:**

The paper lacks "impact statement" section and doesn't claim limitations and potential negative societal impact of their work.

**Strengths And Weaknesses:**

**Strength**:
- RealLife-Bench revolutionizes task modeling through its Hierarchical Topological Projection (HTP) technology, which transforms abstract, high-level semantic instructions into Directed Acyclic Graph (DAG) structures that accommodate a diverse, multi-solution space.
- By transitioning away from static Visual Question Answering (VQA) paradigms, the framework establishes a dynamic, closed-loop evaluation that rigorously assesses a model's ability to ground actions in physical environments through continuous perception-action-reflection cycles.


**Weaknesses**:
- The study lacks human study data, making it difficult to measure the performance gap between AI and humans or verify the reliability of the automated scoring.
- Although HTP is described as a scalable method, its application is restricted to the AI2-THOR environment, leaving its generalization unproven (such as, VirtualHome, ThreeDWorld, Behavior...)
- This is an benchmark paper, but only evaluates only five models and lacks testing on smaller 4B/8B versions, failing to discuss how model scale influences embodied performance.

**Minor Weakness**:
- The fonts in images are excessively small.

---

### Official Review · Reviewer_9k1t · 2026-03-11

**Soundness:** 2
**Presentation:** 1
**Significance:** 2
**Originality:** 2
**Overall Recommendation:** 2
**Confidence:** 4

**Summary:**

This paper introduces RealLife-Bench, which is a benchmark designed to evaluate VLMs on open-ended household tasks inside simulated environments. It creates tasks using scene-agnostic instruction templates and directed graph task structures that are projected to specific environments to generate trajectories and evaluation data. Then, their proposed benchmark includes both end to end interative eflaution and several process level diagnositics like task decomposition, progress management, action prediction, and feedback correction to analyze the abilities of the VLMs based emodied agents.

**Compliance With Llm Reviewing Policy:**

Affirmed.

**Final Justification:**

The authors did not submit the rebuttal, and my concerns are not resolved , so I maintain my original score.

**Key Questions For Authors:**

Please see the weaknesses.

**Limitations:**

The paper is missing the discussion about the limitations and potential negative social impacts of their proposed method.

**Strengths And Weaknesses:**

**Strengths:**
1. The paper has a good problem motivation, since it studies a central VLM-related problem in realistic embodied environments.
2. The introduced benchmark combines both online evaluation with several process level evaalutions, which can potentially provide a fine-grained analysis of model behavior.
3. The scale of the evaluated datasets is moderately sized, which may provide some initial coverage of household tasks.

**Weaknesses:**
1. Many components, such as the hierarchical topological projection and DAG-based scoring system, lack rigorous theoretical justification or ablation studies.
2. Several components rely on automated judging using another language model, which can introduce potential bias in scoring without proper validation of judge consistency.
3. The paper doesn't sufficiently analyze failure cases for the given tasks.
4. The writing quality is inconsistent and sometimes difficult to follow. For instance, the important method details are scattered across sections, and several concepts, like the projection mechanisms and trajectory generation, are described in a very vague way. Also, some figures and tables lack sufficient explanation.
5. The overall novelty and impact of the proposed benchmark are limited, since many existing embodied AI benchmarks already evaluated multimodal agents in a simulated environment for the given task, and the paper's claims of improvements seem incremental.
6. The benchmark is limited to only a single simulation environment and a narrow set of household tasks, which can raise questions about whether it can generalize well to real-world embodied tasks.

---

### Official Review · Reviewer_Gaiz · 2026-03-13

**Soundness:** 2
**Presentation:** 2
**Significance:** 2
**Originality:** 2
**Overall Recommendation:** 2
**Confidence:** 4

**Summary:**

RealLife-Bench is a new benchmark to evaluate VLMs on open-ended household tasks in a simulated environment. The motivation is to move away from very specific task settings and start with more scene-agnostic descriptions of intent (essentially high-level goals) and map those onto more specific task trajectories over a variety of environments. There are both end-to-end and more detailed process-level evaluations, so the goal is not just to evaluate whether a model works or not, but to identify what is holding the model back.

**Compliance With Llm Reviewing Policy:**

Affirmed.

**Final Justification:**

I did not see a rebuttal and I plan to keep my ratings as is.

**Key Questions For Authors:**

- The paper introduces yet another embodied benchmark in an already very crowded space. Could the authors clarify more directly why a new benchmark is needed here, and what concrete gap (instead of listing a long list of design items of what the authors have done) RealLife-Bench fills that is not already addressed by recent embodied task benchmarks?

- The reported performance appears fairly strong for several models. Should readers interpret this as evidence that the benchmark may be relatively easy to saturate, or that it may have limited resolution for distinguishing stronger models? Along the same lines, why is Gemini 3 the only frontier-scale model included in the evaluation?

- The paper describes the evaluation as holistic, but in practice much of the assessment seems to rely on decomposed process checks and VLM-based judging (I checked the appendix). Could the authors clarify what exactly makes the evaluation holistic, and whether any human evaluation was conducted to verify the reliability of these judgments?

**Limitations:**

I didn't see such a section.

**Strengths And Weaknesses:**

## Strengths

- A clear strength of the paper is the benchmark contribution itself. RealLife-Bench covers a reasonably broad set of household activities and scenes, which gives the evaluation more diversity than many narrower embodied benchmarks. I also think the diagnostic aspect is valuable.
- The experiments, although limited in model count, go beyond reporting overall success rates and suggest that instruction understanding, perception, and grounding play different roles in task completion. In particular, the finding that perception and grounding are the main bottlenecks is useful and gives the benchmark some analytical value beyond simple benchmarking.

## Weaknesses

- The claim of realism wasn't convincing to me. The discussion around "real-life" household tasks sounds more convincing than what the actual construction of the benchmark supports. In the end, the tasks are still constructed on top of the predefined ontology of objects and the simulator provided by AI2-THOR. The manually annotated logic of the instructions is still represented as a DAG. This is certainly useful for better controllability. However, it still feels rather artificial. I wasn't convinced that the gap to real-life task execution is actually closed rather than a slightly broader synthetic benchmark, check [4] for example.
- Another aspect that I found lacking is the discussion of related work. The paper justifies itself at least in part by pointing to the limitations of previous benchmarks for embodied AI, such as ALFRED. Yet many of these issues have at least partially been addressed in more recent work [1-3]. I feel that the paper would benefit from a more careful positioning in relation to modern embodied benchmarks and evaluation frameworks. I would have liked to see a discussion of better baselines or comparisons to more well-designed recent tasks such as TEACh, rather than a justification that feels somewhat outdated.

[1] TEACh: Task-driven Embodied Agents that Chat (https://arxiv.org/abs/2110.00534)

[2] SimWorld-Robotics: Synthesizing Photorealistic and Dynamic Urban Environments for Multimodal Robot Navigation and Collaboration (https://arxiv.org/abs/2512.10046)

[3] Virtual Community: An Open World for Humans, Robots, and Society (https://arxiv.org/abs/2508.14893)

[4] Teaching Embodied Reinforcement Learning Agents: Informativeness and Diversity of Language Use (https://aclanthology.org/2024.emnlp-main.237/)

---

### Official Review · Reviewer_Qvma · 2026-03-13

**Soundness:** 2
**Presentation:** 2
**Significance:** 2
**Originality:** 2
**Overall Recommendation:** 2
**Confidence:** 5

**Summary:**

This paper introduces RealLife-Bench, a benchmark to evaluate VLMs in open ended household tasks. The benchmark consists of 15 daily activities across 120 scenes with a total over 1000 tasks and language instructions with different levels of complexity. It constructs an online evaluation pipeline to assess models’ underlying capabilities in task decomposition, progress management, action prediction, perception grounding, and feedback correction. It further reports empirical results from a variety of STOA VLMs on the benchmark.

**Compliance With Llm Reviewing Policy:**

Affirmed.

**Final Justification:**

no rebuttal, keep my score.

**Key Questions For Authors:**

- see weaknesses
- line 220-223 left column, “12”, “13” typo?
- What’s “ON” in figure 1, task completion?
- Notations are inconsistant. E.g., H_t captures different information (line 202, right vs line 244 right)

**Limitations:**

No explicit discussion on limitations.

**Strengths And Weaknesses:**

Strengths:
- A new benchmark for studying instruction following with different levels of abstraction and difficulties.
- The pipeline for process evaluation beyond end task success provides informative assessment of models’ underlying capabilities.

Weaknesses:
- DAG is a center piece for both data generation and planning at inference time. For data generation, how do you ensure the mapping DAG to scene graphs is correct as it will determine the subsequent plans? For inference time planning, it seems the model is only given a positive score if it falls into the boundary that is defined by DAG, otherwise, even the output is valid (although outside of DAG), it would not be counted. This close world setup can be problematic (see below).
- Process evaluations seem tightly coupled with the ground-truth trajectories. However there are many possibilities for “I’m hungry”. A plan derived by a VLM could be a valid plan but not in the ground-truth. how do you evaluate such valid plan?
- There are many similar benchmarks in the AI2Thor environment. While these benchmarks are interesting five years ago, the environment does not model the physics of the world. There is a huge gap from the simulation to the real world. Although this benchmark attempts to address some new angle (e.g., complexity of instructions, process evaluation), it feels incremental. The basic limitations of the simulation such as close-world assumption (in this case DAGs) still remains. It’s not foundamentally different from previous benchmarks.

---

### Decision · Program_Chairs · 2026-04-30

**Decision:**

Reject

**Comment:**

RealLife-Bench is a VLM evaluation framework for open-ended household tasks in AI2-THOR, using scene-agnostic task skeletons projected into scene-specific trajectories via a hierarchical topological mapping over multi-solution DAGs.

All four reviewers rated it below threshold (2/2/2/3). The most confident reviewer raised a fundamental concern: valid plans outside the predefined DAG are marked incorrect, effectively reducing an open-ended task to a constrained multiple-choice setting, and the VLM-as-judge metric lacks human validation. Other reviewers independently noted the reliance on a single simulator, limited model coverage with only one frontier model, absence of a human baseline, inconsistent notation, and unclear descriptions of the core methodology.

The authors did not submit a rebuttal.